# Widespread PREMA in the upper mantle indicated by low-degree basaltic melts

Ronghua Cai[1], Jingao Liu [1] ✉, D. Graham Pearson [2], Andrea Giuliani [3], Peter E. van Keken[4] & Senan Oesch [3]

Studies of ocean island basalts have identified a Prevalent Mantle (PREMA) component as a fundamental feature of mantle geochemical arrays; however, its origin and distribution are highly controversial, including its potential link to plumes sourced in low-shear-wave velocity provinces (LLSVPs) above the core-mantle boundary. In this study, we interrogate the compositional systematics of ~ 3500 Cenozoic oceanic and continental sodic basalts to provide insights into the origin and distribution of PREMA. We find that low-degree basaltic melts with high Nb concentrations located away from deep-mantle plumes have PREMA-like Sr–Nd–Hf isotopic signatures, implying that PREMA is highly fusible and not exclusively associated with LLSVPs. Geochemical modelling and mantle convection simulations indicate that PREMA could have been generated soon after Earth accretion, experiencing only minimal melting or enrichment, and then scattered throughout the upper mantle, rather than being the result of mixing between depleted and enriched mantle components.

Based on the radiogenic-isotope and trace-element characteristics of ocean island basalts (OIBs) and mid-ocean ridge basalts (MORBs), four basic mantle components (depleted MORB mantle: DMM, Enriched Mantle 1 and 2 (EM-1 and EM-2), and high-μ mantle (HIMU)) have been identified, which reflect the large-scale chemical heterogeneity of the asthenosphere[1,2]. A fifth component, known variously as PREMA, FOZO, or C (Table 1), was identified based on the main peak in the frequency distribution of OIB isotopic compositions and exhibits moderately depleted Sr–Nd–Hf isotope compositions lying between DMM and chondrite[1,3,4] (Fig. 1). It was postulated that PREMA, whose composition corresponds to the "Focus Zone" (FOZO) of geochemical trends formed by OIBs from each locality in Sr–Nd–Pb isotopic space, represents a fundamental mantle component located in a boundary layer such as the mantle transition zone or lowermost mantle[4,5] (Fig. 1). The "C" (common component) was also defined by the convergence of oceanic basalts to a common mantle source with moderately depleted Sr–Nd–Hf isotope compositions[3] (Fig. 1). Because PREMA, FOZO and C

have similar isotopic compositions and might represent the same mantle component[6], we treat them together as "PREMA" in the following discussion (Fig. 1). Since the definitions of these PREMA components in previous studies are complex and constantly changing, we have compared previously proposed ranges of PREMA (FOZO and C) in Fig. 1. In the following, we employ a range (Fig. 1) of Sr–Nd–Hf–Pb isotopes for PREMA ($^{87}Sr/^{86}Sr$: 0.703–0.704; $^{143}Nd/^{144}Nd$: 0.5128–0.5130; $^{176}Hf/^{177}Hf$: 0.2829–0.28305; $^{206}Pb/^{204}Pb$: 19–20; $^{207}Pb/^{204}Pb$: 15.55–15.67; $^{208}Pb/^{204}Pb$: 38.8–39.7), which essentially encompass the intervals defined by ref. 5 and ref. 7. Previous studies invoked a primordial origin for PREMA based on the high $^3He/^4He$ of OIBs displaying PREMA-like Sr–Nd–Pb isotope compositions[1,4]. However, other authors have favored the origin of PREMA by mixing depleted and enriched mantle components[7]. Thus, despite its significance in constraining the present and past composition and dynamics of Earth's mantle, the origin of PREMA and its distribution in the mantle remain unclear.

[1]State Key Laboratory of Geological Processes and Mineral Resources, China University of Geosciences, Beijing 100083, China. [2]Department of Earth and Atmospheric Sciences, University of Alberta, Edmonton, Alberta T6G 2E3, Canada. [3]Institute of Geochemistry and Petrology, Department of Earth Sciences, ETH Zurich, 8092 Zurich, Switzerland. [4]Earth and Planets Laboratory, Carnegie Institution for Science, Washington, DC 20015, USA. ✉ e-mail: jingao@cugb.edu.cn

**Table 1 | The comparison between the classic "PREMA" and "PREMA" focused on in this study**

| | Sr–Nd–Hf–Pb isotopes | $^3$He/$^4$He | Sampled by what kind of mantle melts | Geological distribution | Physical property | Origin |
|---|---|---|---|---|---|---|
| Classic PREMA | Moderately depleted Sr–Nd–Hf isotopes and radiogenic Pb isotopes | Highly variable, and no clear relationship with Sr–Nd–Hf isotopes | Tholeiitic OIB, low- to medium Nb basalts | Edges of LLSVPs | Unknown | 1) Primitive production of early Earth differentiation 2) Mixture of DM and EM |
| PREMA focused on in this study | Moderately depleted Sr–Nd–Hf isotopes and radiogenic Pb isotopes | Data scarcity, requirement of further studies | Kimberlites, high-Nb basalts (alkali, silica-undersaturated basalts) | All geological settings, mainly in the thick lithosphere (continents and mature oceanic crust) | Highly fusible | Likely early-depleted mantle, widespread in Earth's upper mantle |

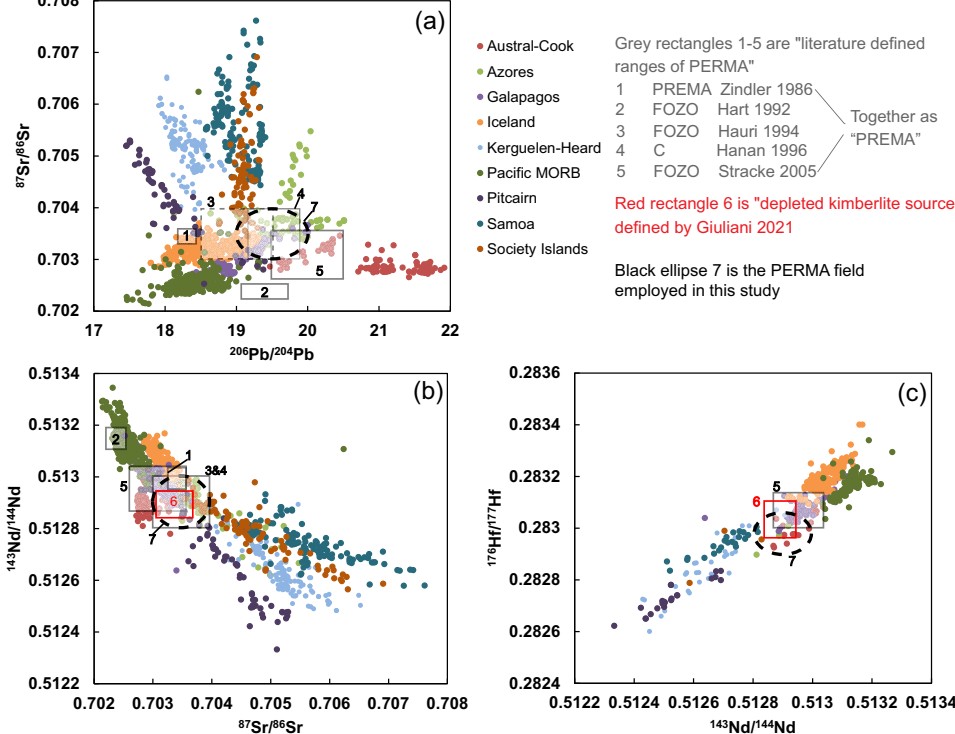

**Fig. 1 | Present-day Sr–Nd–Hf–Pb isotope compositions of MORBs and OIBs.** In the plots **a**–**c**, the grey rectangles represent the range of "PREMA" defined in the literature refs. 1,3–5,7 while the black ellipses indicate the "PREMA" field employed in this study ($^{87}$Sr/$^{86}$Sr: 0.703-0.704; $^{143}$Nd/$^{144}$Nd: 0.5128-0.5130; $^{176}$Hf/$^{177}$Hf: 0.2829-0.28305; $^{206}$Pb/$^{204}$Pb: 19-20; $^{207}$Pb/$^{204}$Pb: 15.55-15.67; $^{208}$Pb/$^{204}$Pb: 38.8-39.7). The red rectangles represent the range of "depleted kimberlite source"[9], while kimberlites do not have well-defined ranges in Pb isotope compositions hence preventing an estimation of the kimberlite-based PREMA for Pb isotopes. The data of MORBs and OIBs are from ref. 55. The plots of $^{208}$Pb/$^{204}$Pb vs $^{207}$Pb/$^{204}$Pb and $^{206}$Pb/$^{204}$Pb are shown in Fig. S8.

The temporal evolution of Nd–Hf isotopes in kimberlites has provided a perspective on the temporal evolution of mantle heterogeneities, including the origin of the PREMA-like component[8,9]. Volatile-rich, small-volume kimberlites only occur within cratons (Fig. 2) and are regarded as the deepest mantle-derived melts[10–12]. Fresh kimberlites consistently tap a source with a relatively homogeneous Nd–Hf isotope composition that is intermediate between that of chondrite and DMM ("depleted kimberlite source") and is indistinguishable from PREMA (Fig. 1), suggesting the preservation of a PREMA-like component for at least the past ~2 Gyr[9]. Hence, PREMA might represent a long-lived, minimally processed mantle component potentially derived from early differentiation of chondritic mantle[9].

To further address the origin and distribution of the PREMA-like component in the asthenosphere and its relationship with kimberlites, we have examined an extensive compilation of bulk-rock compositional data for global Cenozoic sodic 'basalts' ranging from alkali-basalts to more silica-undersaturated melilitites, nephelinites, and basanites (Fig. S1a). These magmas were selected because those more enriched in highly incompatible elements (e.g., Nb-Ta-La) share similar trace-element patterns to kimberlites[13,14]. These observations point to low-degree partial melting of similar mantle sources at depths of 50–125 km to those that generate kimberlites[15,16]. The signature of fusible mantle components becomes diluted during high-degree melting, making small-degree melts more attractive to examining these components. Moreover, the key trace element signatures of such incompatible element-rich low-degree melts are not easily affected by crustal contamination, also providing a clearer view of the mantle signature than large-degree mantle melts with lower incompatible element contents. In contrast to kimberlites, sodic basalts are emplaced in different tectonic settings and sample much broader regions of the upper mantle, including ocean basins and regions of thinner continental lithosphere[17] (<125 km[16]) not affected by deep mantle upwellings (Fig. 2) and thus provide a more comprehensive picture of geochemical variations in Earth's mantle.

## Results and discussion

### Moderately depleted components sampled by global high-Nb basalts

The radiogenic isotopes of global sodic silica-undersaturated basalts vary as a function of partial melting degree as shown by relationships

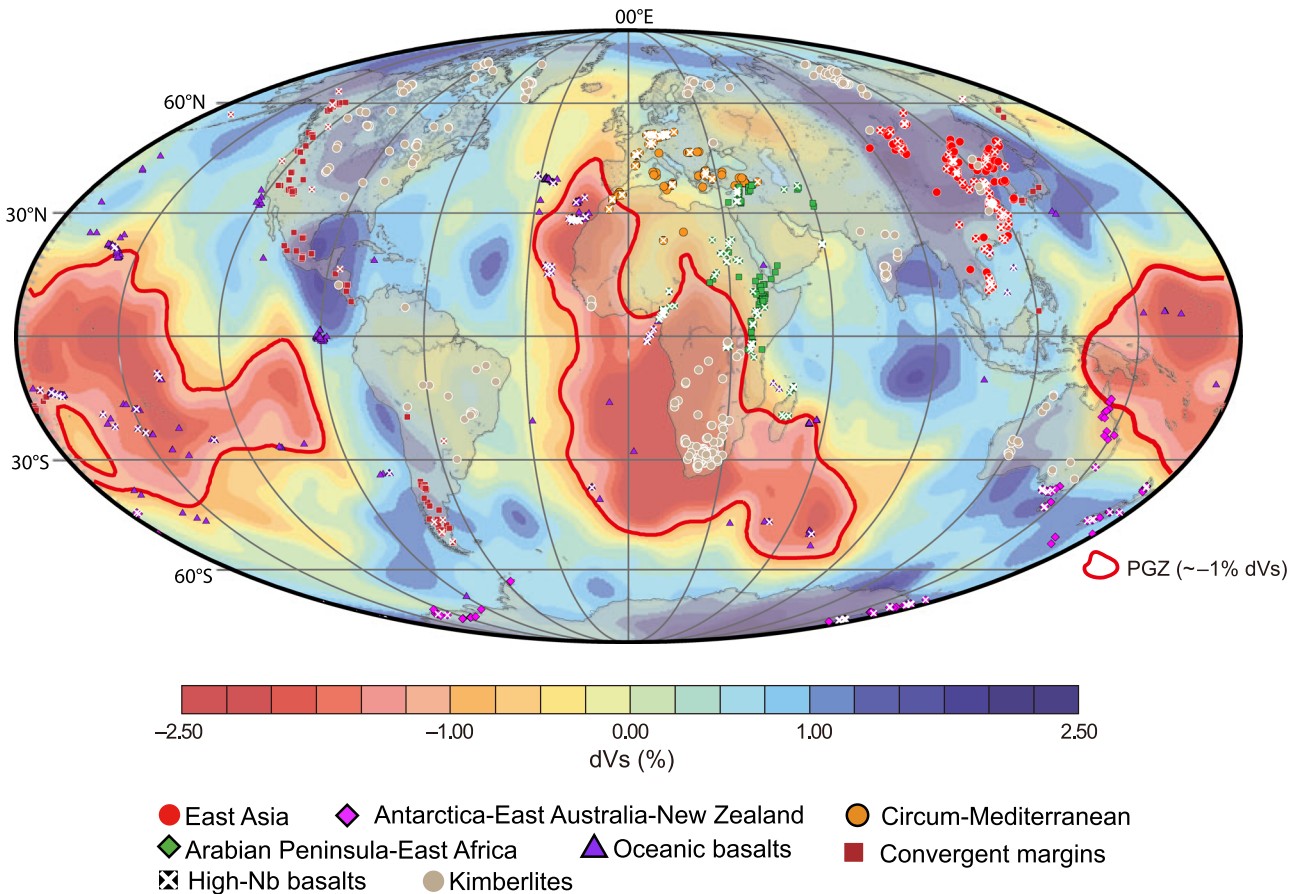

**Fig. 2 | World map showing the global distribution of Cenozoic sodic basalts.** These rocks are categorized into six groups based on their geographic distribution. The symbols covered by a white cross represent high-Nb basalts (Nb-N > 100, where Nb-N indicates Nb contents normalized to the primitive mantle composition[18]). The shear wave velocity map at the core–mantle boundary (2800 km) is from ref. 56 (this figure is modified from ref. 57); the thick red lines are interpreted to be the plume generation zones (PGZ) at the edges of the two large low shear-wave velocity provinces (LLSVPs).

with the concentrations of highly incompatible elements, including Nb, Ta, U, Th, and La (Figs. S2, S3, and S9). Niobium content (Fig. 3) is here employed as a proxy of the degree of mantle melting because Nb is relatively immobile in hydrous fluids and hence robust to alteration after magma emplacement. Covariation diagrams between Nb and radiogenic isotope ratios or trace element ratios reveal that there are no significant differences between basalts from oceanic vs. continental settings (Figs. 3 and S9), which justifies the employment of continental basalts to examine geochemical variations in the upper convecting mantle. After binning the data using a bin width of 50 Nb-N units (where Nb-N indicates Nb contents normalized to the primitive mantle composition[18]), global sodic basalts can be divided into four groups based on their covariations between Nb-N and radiogenic isotopes (Fig. 4). The covariation trends between oceanic and continental basalts are very similar (Fig. 4). "Low-Nb" (Nb-N < 50) and "medium-Nb basalts" (50 < Nb-N < 100) from both oceanic and continental settings show relatively large variations in Sr-Nd-Hf isotopic compositions (Fig. 4), with a high proportion of values that correspond to geochemically 'enriched' compositions (i.e., high $^{87}Sr/^{86}Sr$, low $^{143}Nd/^{144}Nd$). In contrast, "high-Nb" (100 < Nb-N < 150) and especially "very high-Nb basalts" (150 < Nb-N < 300) exhibit a much more restricted range for all the isotopes considered (Fig. 4). The variability in isotopic composition, estimated by median absolute deviations (MADs) of Sr-Nd-Hf isotope ratios, is typically the highest for "medium-Nb basalts" and reaches minimum values in the "very-high Nb basalts" at Nb-N values between ~150 and 200 (Fig. S4), i.e., those with compositions closest to PREMA show the least variability.

High-Nb basalts from oceanic settings have a more homogeneous and marginally more depleted composition compared to continental basalts (Fig. 3). Although we have filtered the data using multiple indices of crustal contamination (see details in Methods), the greater isotopic diversity and elevated $SiO_2$ contents of continental sodic basalts compared to oceanic sodic basalts (Fig. 3a–d, h) potentially suggest some contribution from continental lithosphere in the continental variants. However, continental and oceanic sodic basalts converge to similar Sr-Nd-Hf-Pb isotope ratios and elemental ratios (e.g., Nb/Th, Ce/Pb, Ti/Eu) with increasing Nb contents (Fig. 3). This observation indicates the tapping of broadly similar convecting mantle sources by both continental and oceanic basalts with high and very high Nb contents, regardless of the potential for unidentified crustal contamination in some of the continental basalts.

Classification of samples in our dataset using the total alkali versus silica approach[19] reveals that the highest Nb concentrations are generally associated with the most alkaline and silica-undersaturated compositions such as basanites and melilitites (Fig. S1a). In addition to having a restricted isotopic range, high-Nb basalts from different geological settings have remarkably similar trace element patterns (Fig. 5a). Ratios of trace elements with similar incompatibility during partial melting (e.g., Nb/Th, Ce/Pb, Ba/Th; Fig. 3) show no significant correlations with Nb-N. In contrast, Zr/Nd, Ti/Eu, Ti/Gd, and silica are negatively correlated with Nb-N (Fig. 3). The decrease of Zr/Nd (Fig. S5) is largely consistent with higher compatibility of Zr in the mantle residue compared to Nd (Fig. 5a). Conversely, the coupled decrease of Ti/Eu and Ti/Gd might suggest the increasing involvement of a carbonate component[20] in the source of sodic basalts with higher

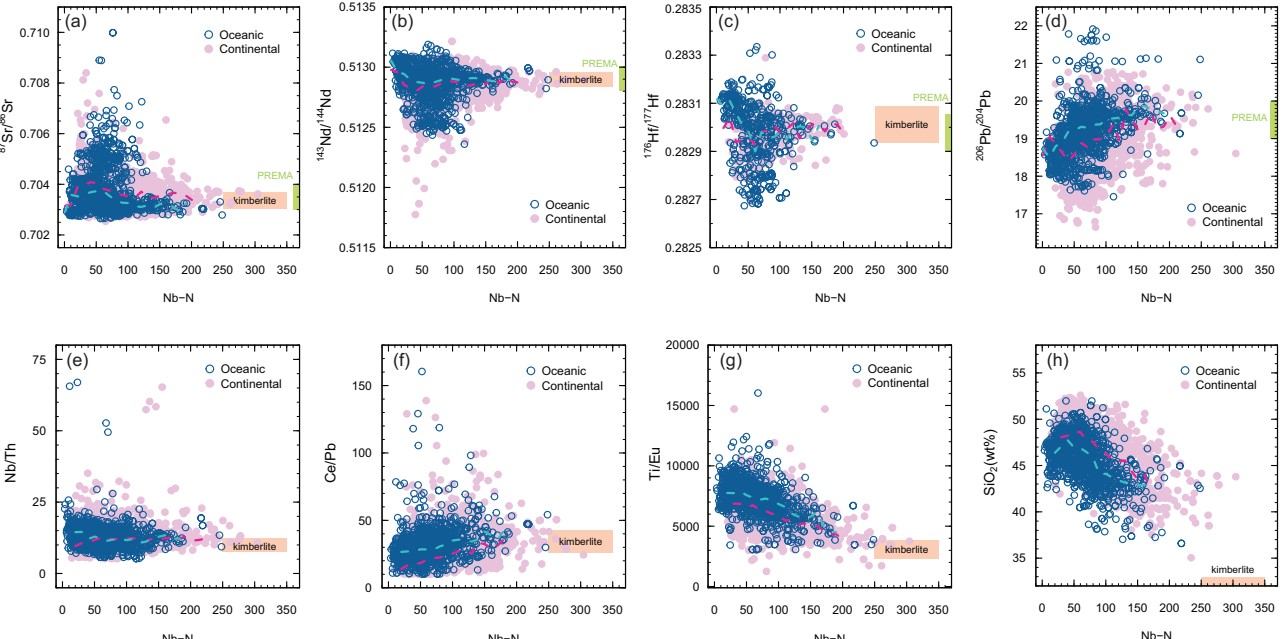

**Fig. 3 | Covariations of key geochemical proxies for global Cenozoic sodic basalts.** Plots of Nb–N vs **a**–**d** radiogenic isotope ratios, **e**–**g** incompatible trace element ratios, and **h** silica for Cenozoic sodic basalts from continental and oceanic domains. Nb–N represents Nb concentrations normalized to the primitive mantle value of ref. 18. The dotted curves show median values for continental (red) and oceanic (light blue) sodic basalts based on a binning width of 5 Nb–N units. The kimberlite field is based on the trace element composition of primary kimberlite melts in ref. 58 for major and trace elements and the depleted kimberlite source of ref. 9 for radiogenic isotopes. The PREMA bar is taken from the range proposed by this study (Fig. 1).

Nb contents (Fig. 3g). This hypothesis is entirely consistent with decreasing $SiO_2$ contents, and hence the extent of Si under-saturation, with increasing Nb contents (Fig. 3h).

These observations combined suggest that silica-undersaturated sodic basalts generated by low degrees of partial melting (yielding high Nb–N) sample a relatively more homogeneous mantle source with moderately depleted Sr–Nd–Hf isotope compositions compared to medium- and low-Nb basalts that access a greater range of source compositions by larger degrees of melting (Fig. 4).

### A PREMA-like component sampled by both kimberlites and high-Nb basalts

The present-day Sr–Nd–Hf isotope compositions of the "depleted kimberlite source"[9] and PREMA (excluding Pb isotopes, as kimberlites do not have well-defined ranges in Pb isotope compositions hence preventing estimation of the kimberlite-based PREMA for Pb isotopes) overlap the compositional range of high-Nb sodic basalts (Figs. 1, 3, and 4). Interestingly, global kimberlites have similar incompatible element ratios (Fig. 3) and primitive-mantle normalized trace element patterns as high-Nb basalts (Fig. 5a). Compared to high-Nb basalts, kimberlites have higher concentrations of highly incompatible elements, lower heavy rare earth element (HREE) contents and more pronounced negative anomalies of K–Zr–Hf–Ti (Fig. 5), features consistent with lower melting degrees and deeper melting depths for kimberlites compared to high-Nb basalts. The analogous trace element patterns documented in global kimberlites have been attributed to the tapping of similar sources or equilibration with common mantle components[21].

Two models are commonly associated with the formation of silica-undersaturated alkaline basalts: (1) low-degree partial melting of metasomatized lithospheric mantle in the presence of phlogopite or amphibole[22]; and (2) melting of carbonated peridotite or eclogite in the asthenosphere[15,16,23–25]. Melting of amphibole-bearing lithospheric mantle cannot explain the similarity in trace element patterns between high-Nb basalts and kimberlites, which are well known to derive from amphibole-free carbon-bearing asthenosphere[26,27]. Instead, the high CaO, low $SiO_2$ contents, and high $CaO/Al_2O_3$ of the high-Nb basalts (Fig. S6) are consistent with partial melting experiments of carbonated peridotite[15], as also invoked for kimberlites[25]. Petrological modeling using trace-element partition coefficients between peridotite and carbonate-rich melt[28] demonstrates that the trace element patterns of kimberlites can be produced by lower-degree melting at higher pressure (i.e., higher proportions of residual garnet; see Methods) of the same source of high-Nb basalts (Fig. 5b). Furthermore, high-Nb basalts with PREMA-like isotopic compositions occur in both oceanic and continental settings (Fig. 4), which negates an exclusive link to continental lithosphere. We recognize that the elemental compositions of some continental sodic basalts may require derivation—in part or in whole—from a lithospheric source region[29]. In such cases, many basalts retain their PREMA-like isotopic compositions because they are dominated by the signature of asthenosphere-derived metasomatic melts that transferred their compositions to the basal lithosphere shortly before magma genesis[29]. Here, we propose that the similarity of trace elements and radiogenic isotopes between kimberlites and silica-undersaturated high-Nb basalts reflects derivation from common carbon-bearing mantle sources containing fusible PREMA-like components at differing depths.

### Origin and distribution of the PREMA-like component

The apparent spatial coincidence of the paleo-geographic locations of most Phanerozoic kimberlites[11] and high $^3He/^4He$ OIBs[30] with plume generation zones at the margins of LLSVPs (Fig. 2) might support the notion that the PREMA-like component[9] may reside in LLSVPs[30–32]. In contrast, Cenozoic high-Nb basalts with PREMA-like isotopic signatures occur in various geological settings—both continental and oceanic—with many located far away from LLSVPs at the time of their eruption. Therefore, plumes stemming from the LLSVPs are not the only possible source of PREMA (Fig. 2). From our analysis, it appears that PREMA represents a ubiquitous feature of the convecting mantle, including upper mantle sources.

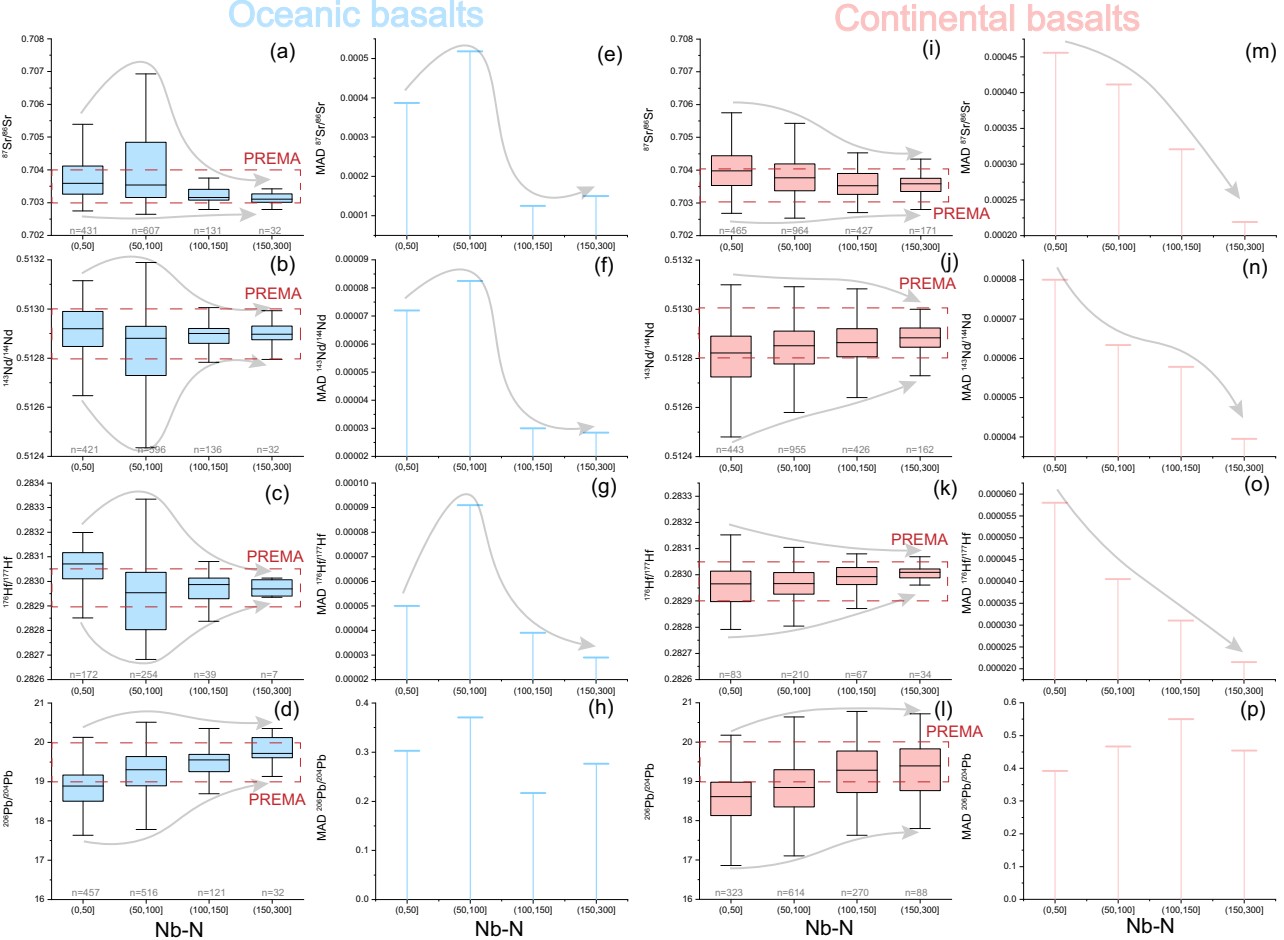

**Fig. 4 | Similar covariation trends between oceanic and continental basalts.** **a**–**d**, **i**–**l** Box and whisker plots and **e**–**h**, **m**–**p** median absolute deviations (MAD) of the Sr–Nd–Hf–Pb isotope compositions of global Cenozoic sodic basalts divided into four groups based on Nb–N. Nb–N represents Nb contents normalized to the primitive mantle[18]. The "n = X" represents the sample number. The box defines the 25th–75th percentiles, with the line in the box marking the medium value and the whiskers standing for 0th to 100th percentiles, excluding outliers.

Preferential sampling of PREMA by low-degree melts suggests a lower melting temperature of PREMA compared to other mantle components, including enriched mantle. The highly fusible nature of PREMA may be associated with the presence of $CO_2$ during melting because carbonated peridotite has a lower solidus than other components, including pyroxenites[33]. However, the carbon associated with PREMA is unlikely to contain significant recycled crustal material because any exotic C-bearing fluid will also introduce other incompatible elements, such as Sr and Nd, hence perturbing the geochemically depleted isotopic signature of PREMA[34]. Oxidation of ancient reduced carbon by redox melting[35,36] may be important to reconcile the presence of oxidized carbon in the PREMA component with the lack of recycled crustal material.

The generation of the spectrum of mantle melts ranging from kimberlite, nephelinite, and basanite to alkali-basalt is, in large part, a function of lithospheric thickness[16,17,37] because the thickness of the lithospheric lid controls the degree of melting. A correlation between lithospheric thickness and indices of the extent of melting has been observed for intraplate magmatism both regionally, e.g., in eastern China[38], and globally in OIBs[37]. High-Nb basalts resulting from low degrees of melting are indeed more widespread in continental lithosphere (24% of all the sodic basalts) than comparatively thinner oceanic lithosphere (7% of all the sodic basalts), while kimberlites erupting through the thickest lithosphere associated with cratons have higher Nb contents than high-Nb basalts (Fig. S1b). Hence, the PREMA component is preferentially sampled in a purer form by high-Nb basalts and kimberlites that form beneath thick lithospheric lids, making examination of continental basalts a valuable and under-exploited tool to explore mantle geochemistry. The greater compositional diversity of medium- and low-Nb basalts (Figs. 3 and 4) that are derived from shallower depths beneath the thinner lithosphere, can be attributed to an increasing contribution to the melting products from geochemically enriched components (e.g., garnet pyroxenites with initial partial melting at ~100 km[39–41]) and other more refractory components.

The PREMA-like component sampled by low-degree basaltic melts cannot be the result of mixing between depleted and enriched mantle components for two reasons: First, low-degree melting of a mixed source will produce melts with geochemically enriched compositions which are not observed among the silica-undersaturated high-Nb basalts. Second, mixing between DMM (i.e., MORB-like) and EM components does not intersect the field of PREMA-like high-Nb basalts in plots of incompatible trace element ratios (e.g., Ce/Pb, Nb/Th) versus Sr, Nd and Hf isotopes (Fig. 6). In other words, PREMA-like high-Nb basalts cannot be formed by mixing melts derived from DMM and EM sources. Additionally, we find that high-Nb basalts do not seem to be clearly related to the melting of pyroxenite sources when compared to sodic basalts with lower Nb contents (Fig. S7).

The isotopic record of kimberlites as old as 2.1 Ga seems to support the notion that PREMA may be a long-lived, relatively

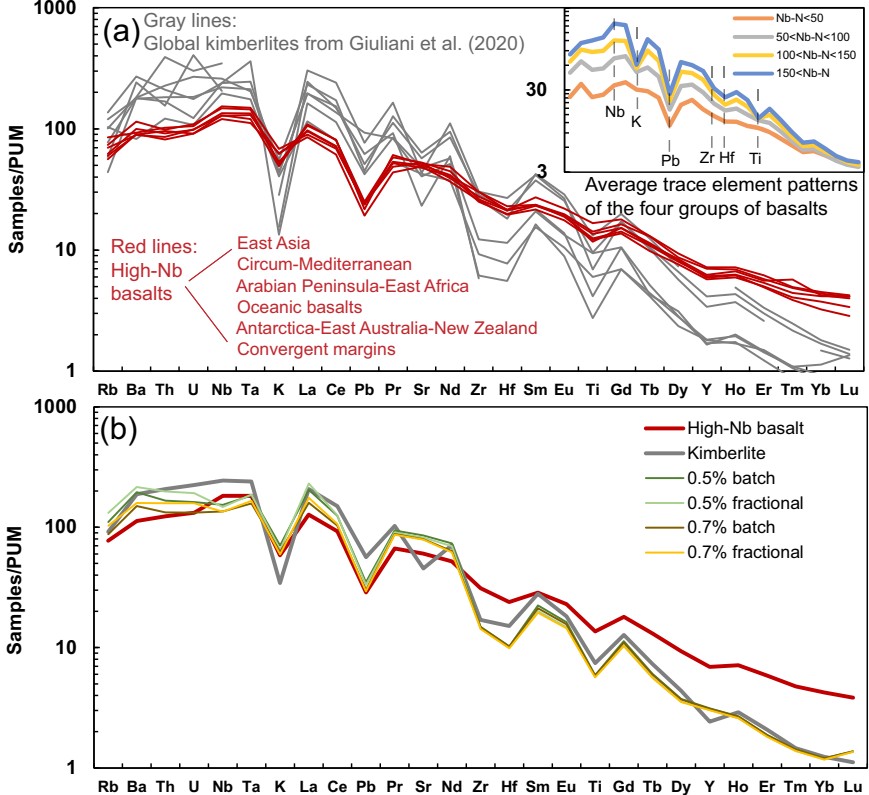

**Fig. 5 | Primitive mantle-normalized trace element patterns of sodic basalts and kimberlites. a** Kimberlites (grey lines) and high-Nb basalts (Nb−N > 100; red lines) from different localities worldwide; kimberlite data from ref. 21. The inset shows the average trace element patterns of the four groups of basalts. **b** Results of the partial melting model plotted along with the average values of kimberlites and high-Nb basalts. Primitive mantle values are from ref. 18.

unprocessed mantle component, potentially dominated by material derived from early Earth differentiation[9]. This hypothesis is strengthened by the short-lived $^{182}$Hf–$^{182}$W isotopic data in kimberlites, which show that geochemically depleted kimberlites have modestly negative $\mu^{182}$W values (−5.9 ± 3.6 ppm) consistent with an ancient origin[42]. To explore the feasibility of an ancient origin of PREMA and its widespread distribution in the convecting mantle, we have re-examined the results of recent simulations of whole-mantle convection spanning the age of Earth[43,44] (see Methods). We assume that the unprocessed mantle derives from early differentiation of chondritic Earth and, therefore, has PREMA-like composition[9] in broad agreement with a super-chondritic model[45–47]. This model shows that slabs sink into the lower mantle and displace minimally processed PREMA-like material upwards. As a result, the most primitive material is not concentrated solely in the lower mantle but instead becomes distributed throughout the whole mantle and widely scattered within the upper convecting mantle (Fig. 7; Supplementary Movie 1). The resulting widespread distribution of PREMA is consistent with sampling of ancient PREMA-like material by continental basalts far away from deep mantle plumes (Fig. 2). In contrast to the well-mixed upper mantle with low viscosity, the model shows that the lower mantle is poorly mixed with unprocessed material or minimally processed tending to congregate at boundary-layers (Fig. 7). Survival of minimally processed and yet fusible PREMA in the convecting mantle for billions of years is consistent with inefficient cycling of mantle material through low-pressure melting zones (Supplementary Movie 1) as well as limited melting of carbon-bearing peridotite at the prevailing reducing conditions that occur below the metal saturation depth (-250 km)[36].

To summarize, our global survey of Cenozoic alkaline basalt compositions, integrated with mantle convection simulations, provides hints that PREMA is a physically discrete, highly fusible constituent that is widespread in Earth's upper mantle, rather than being exclusively associated with deep thermochemical structures such as LLSVPs. The PREMA mantle component is much more prevalent than previously thought—as befits its name—and points to the possible preservation of minimally processed early-Earth heterogeneities throughout the mantle, including the upper mantle, where they can be readily sampled in small-degree melts. This finding bears fundamental implications for the physical evolution of Earth's convecting mantle because it requires lower time-integrated convective vigor, and hence higher viscosity, to allow preservation of ancient and potentially highly fusible geochemical heterogeneities in the upper mantle.

## Methods
### Data compilation and filtering
All geochemical data for mafic igneous rocks used in this study were taken from Precompiled Files in the GEOROC database in August 2021 (http://georoc.mpch-mainz.gwdg.de/georoc/): basalt part1–9, basanite, nephelinite, melilitite, hawaiite, ankaramite. Subalkali basalts were removed from the dataset by employing the classification criteria of ref. 48. Major element data was recalculated to 100 wt.% on a volatile-free basis after removing analyses with a loss on ignition (LOI) outside a −2 to 3 wt% range and only considering analyses with totals (including LOI) between 97 and 102 wt%. Only Cenozoic samples with complete major- and trace element and Sr-Nd isotope compositions were selected. Samples with negative Nb−Ta anomalies, likely due to melting in arc environments or continental crust assimilation, were excluded as they are unlikely to reflect the characteristics of mantle sources unmodified by recent crustal interaction. Hence, we have excluded the samples with Nb/U < 20, Nb/Th < 5, and Ce/Pb < 10 from our dataset. For the same reason, samples from convergent margins

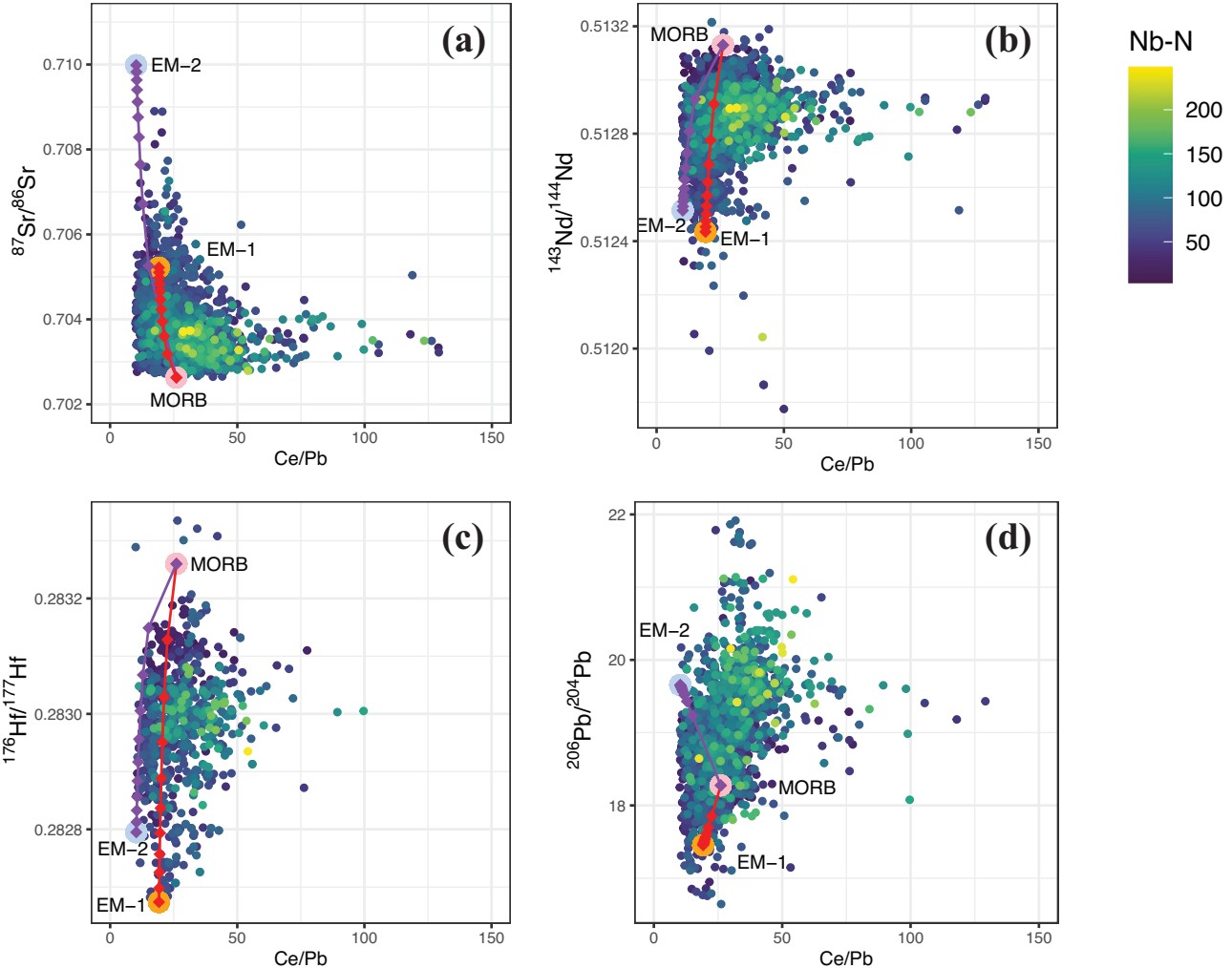

**Fig. 6 | Covariation diagrams of Ce/Pb vs radiogenic isotope ratios for global Cenozoic sodic basalts.** The values of Nb-N are color coded (**a**–**d**). Mixing curves between mid-ocean ridge basalt (MORB) and enriched mantle components (EM-1 and EM-2) are included with each diamond indicating a 10% mixing step. For the MORB endmember, we used trace element concentrations from the average MORB of ref. 59 and isotope ratios for the depleted MORB mantle (DMM) from ref. 60. Trace element concentrations of the EM-1 and EM-2 basaltic endmembers were selected from our screened database, using the samples with the most radiogenic Sr isotope composition from the Pitcairn-Gambier Chain (EM-1) and Samoan Islands (EM-2), respectively. Isotope ratios of the EM components were assumed to be equivalent to the most radiogenic $^{87}$Sr/$^{86}$Sr and the most unradiogenic $^{143}$Nd/$^{144}$Nd, $^{176}$Hf/$^{177}$Hf and $^{206}$Pb/$^{204}$Pb (but most radiogenic $^{206}$Pb/$^{204}$Pb for EM2) composition from the same sample group (i.e., Pitcairn–Gambier and Samoa), respectively.

with low incompatible element contents (i.e., Nb–N < 20; where Nb–N indicates niobium contents normalized to primitive mantle values from ref. 18) were also removed. In addition, potassic basalts (K$_2$O/Na$_2$O > 1) were discarded from this compilation because they are attributed to the melting of (or interaction with) enriched lithospheric mantle and have distinct chemical compositions compared with sodic basalts[49]. To mitigate the geochemical effects of fractional crystallization only melts containing more than 6 wt% MgO were considered. This filtering approach produced a final dataset containing 3453 whole-rock compositions of sodic basalts of the Cenozoic age (ocean: 1333; continent: 2120). The filtered dataset (Supplementary Data 1) contains some samples (n = 269) from convergent margins (e.g., Andean arc; Fig. 2). We strongly emphasize that these samples are not associated with the melting of the arc mantle wedge, instead, they were derived from the shallow asthenosphere not affected by slab fluids or similar material (e.g., slab window[50,51]). Furthermore, the high-Nb basalts from convergent margins (Fig. 5) share similar characteristics with samples from other geological backgrounds and were therefore included in the final compilation; the result does not change if these data are excluded for consideration.

## Partial melting models

We follow the method described in ref. 52 to test whether the trace element patterns of kimberlites can be produced by lower-degree partial melting of a high-Nb basalt source. At first, we calculated the trace element contents of a high-Nb basalt source using modal batch melting (1) or fractional melting (2), where $C_0$ is the source composition, $C_l$ is the basalt composition, and $D$ is the bulk partition coefficient. The element concentrations in the mantle source of high-Nb basalts are calculated with a partial melting degree of 1%.

$$C_0 = C_l[D + F(1 - D)] \quad (1)$$

$$C_0 = \frac{C_l F}{[1 - (1 - F)^{1/D}]} \quad (2)$$

The calculated source compositions are then modeled to melt at lower degrees using the same equations. The mass fractions of minerals are taken from ref. 28. The partition coefficients are from refs. 28,53. If a partition coefficient for an element is not listed (Pr, Tb, Ho, Tm), it is taken as the average of its neighboring elements.

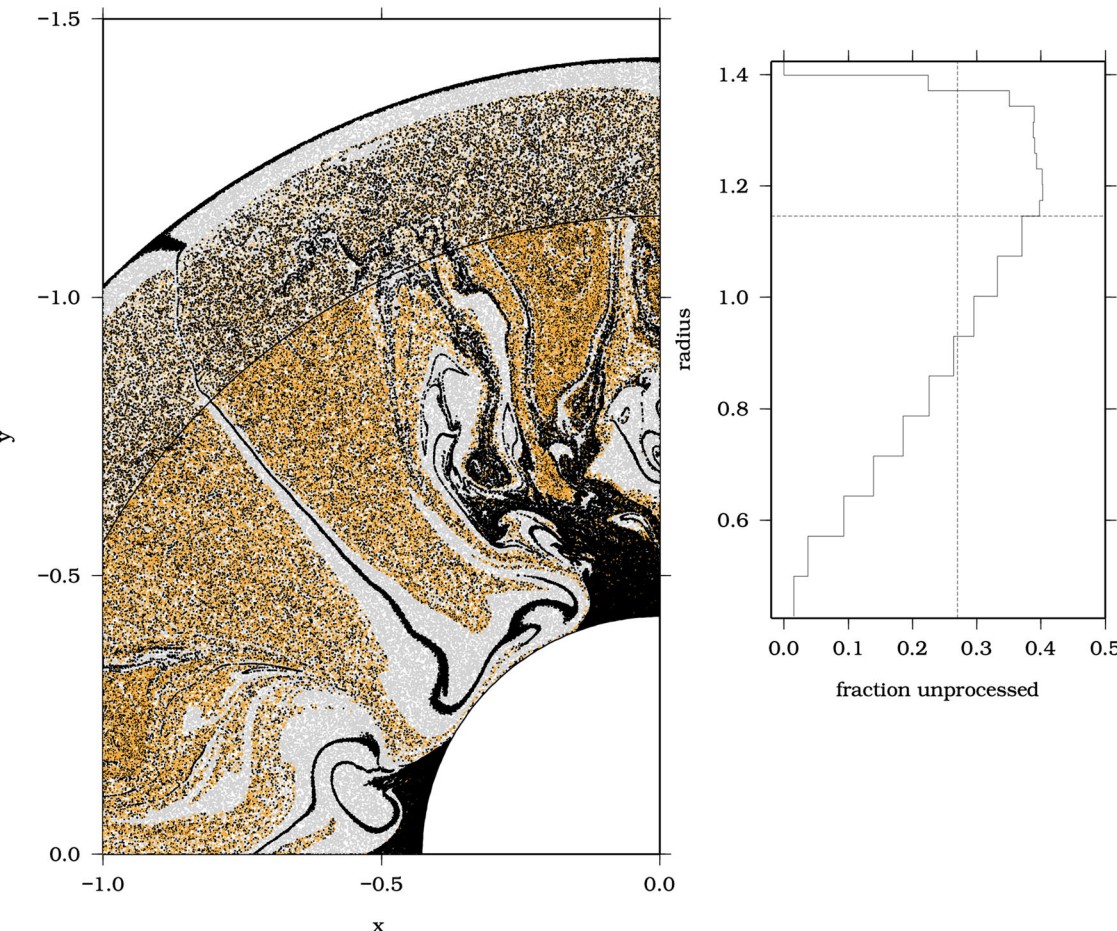

**Fig. 7 | Compositional mantle layering predicted by a global-scale mantle-convection model after 4.57 Ga model time from ref. 31.** Oceanic crust is shown in black, the depleted mantle from which the crust is extracted is shown in grey, and the unprocessed mantle is in orange. Each boundary from outside to inside represents the surface, lithosphere-asthenosphere boundary, 670 km discontinuity, and core-mantle boundary. The diagram on the right shows the present-day depth distribution of the fraction of unprocessed mantle, while the vertical line represents the average fraction of unprocessed mantle.

The initial melt compositions of high-Nb basalt are taken from the average values of global samples. Because the HREE contents of kimberlite are low, our results reveal that the kimberlite patterns can be produced when the ratio of source garnet of kimberlite to high-Nb basalt is high (~5–6) with a partial melting degree of 0.5–0.7%.

## Numerical modeling
Figure 7 is based on a reinterpretation of a model with 6.4% excess density for the subduction oceanic crust, as introduced in ref. 31 and used in ref. 54. No new dynamical modeling was performed as part of this study.

## Reporting summary
Further information on research design is available in the Nature Portfolio Reporting Summary linked to this article.

## Data availability
All geochemical data for mafic igneous rocks used in this study were taken from Precompiled Files in the GEOROC database in August 2021 (http://georoc.mpch-mainz.gwdg.de/georoc/). The final database used in the discussion is provided in the Supplementary Data files.

## Code availability
The computer code for geochemical modeling that is used in this study can be accessed by contacting the authors.

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

## Acknowledgements

This research was financially supported by funding from the National Key R&D Program of China (2022YFF0801004, 2019YFA0708400, and 2020YFA0714800), the National Natural Science Foundation of China (No. 42121002). James Scott, Ruohan Gao, and Yujian Wang are thanked for reading an early version of this manuscript. This is CUGB petrogeochemical contribution No. PGC2015-110 (RIG-No. 27).

## Author contributions

J.G.L. and R.H.C. conceived and designed the study. R.H.C., J.G.L., D.G.P., A.G., and S.O. interpreted the data, developed the hypotheses, and wrote the manuscript. P.E.v.K reinterpreted existing numerical models. All the authors contributed to interpreting the data and writing the paper.

## Competing interests

The authors declare no competing interests.
