## [Peer Review File · Nature Communications]

REVIEWER COMMENTS

Reviewer #1 (Remarks to the Author):

Dear Editor,

I had the pleasure of reading the manuscript by Cai et al., titled "Widespread PREMA in the Upper Mantle Indicated by Low-Degree Basaltic Melts", and I am genuinely excited by the work they have done. I believe they've made significant strides in their research, and their manuscript provides an insightful reading experience. However, I do have a few suggestions that could further enhance the clarity and depth of their study.

1. The authors have admirably updated the isotopic composition of PREMA. They've interestingly used Sr-Nd-Hf isotopes and have reported the $^{206}\text{Pb}/^{204}\text{Pb}$ value. However, I found myself wondering (a) why the values for $^{207}\text{Pb}/^{204}\text{Pb}$ and $^{208}\text{Pb}/^{204}\text{Pb}$ were not provided, and (b) whether Pb was actually utilized, or was it simply extrapolated from Sr-Nd-Hf. Given the focus on PREMA in this study, a detailed discussion on these points could provide valuable insights and clarify these aspects.

2. Their rationale for using continental basalts (lines 76–79) could benefit from a bit more explanation. For instance, it would be great to understand why basalts need to be enriched in highly incompatible elements, and why a low degree of partial melting is preferable over a high degree. More elaborate reasoning here would undoubtedly strengthen their arguments.

3. The study opens up some intriguing questions about potential issues with continental basalts. The possibility of continental crust contamination is an aspect that could be examined, and it's not currently discussed in the manuscript. Moreover, it would be enlightening to explore the possibility of continental basalts being generated in the lithospheric mantle, as inferred by the author at $<100\text{km}$. Although these queries are addressed, they come a bit later in the manuscript and addressing them sooner could provide a smoother flow of ideas.

4. Upon reviewing the provided data and engaging with the content, I am leaning towards the idea that the PREMA signature could potentially stem from the lithosphere itself. This is a fascinating possibility that deserves further exploration.

The authors have indeed done a commendable job, and these suggestions are merely intended to help them enhance their work. I eagerly look forward to seeing the next iteration of their manuscript.

Reviewer #2 (Remarks to the Author):

This study addresses the origin and distribution of PREMA, by examining 3500 Cenozoic sodic continental and oceanic basalts. The authors show high-Nb basalts (equivalent to low degree of melts) tend to preferentially tap PREMA. The main conclusions of this study are: (a) The high-Nb basalts share a common source with kimberlites, being slightly higher degree of partial melts. (b) The PREMA does not result of mixing between depleted and enriched mantle components. (c) The PREMA is a physically discrete, highly fusible constituent that is widespread in Earth's upper mantle, rather than being exclusively associated with deep thermochemical structures such as LLSVPs. While this work is a potential contribution to NC, it has a number of weaknesses/problems that must be addressed before further consideration.

(a) High-Nb basalts are interpreted as low-degree melts, as such they are often alkalic in nature through and are associated with low Si content. In principle such melts are more abundant in continental setting than in oceanic setting, due to the lithospheric lid effect. While this is claimed in the text by quoting the proportion of 24% versus 7%, Fig. 1 does not show compelling evidence. For instance, Nb concentrations in the lavas from two settings are essentially similar. The continental basalts tend to be higher in Si at given Nb, which is the reverse of what is expected. In addition, the continental high-Nb basalts tend to have higher Sr but lower Nd isotopic ratios than oceanic counterparts (looking at the medium curves in Fig. 1 and individual data points). I am thus wondering whether they come from different sources and/or some assimilation processes were involved. Or this is related to the data processing which average out the compositional heterogeneity.

(b) Broadly similar trace element pattern is emphasized between high-Nb basalts and kimberlites. However, high-Nb basalts lack negative anomalies in Sr, Zr and Hf which are conspicuous in Kimberlites. I am not convinced by an interpretation invoking different degree of partial melting. In this regard, a detail evaluation of source lithology is absolutely needed.

(c) Classic PREMA is generally associated with high $^3\text{He}/^4\text{He}$ ratios which is considered as an intrinsic feature of PREMA. The so-called wide-spread PREMA in the upper mantle, as argued in this study, do not show this primordial He isotope composition. This observation may simply reflect different composition between the upper and lower mantle, arguing against widespread PREMA in the upper mantle.

(d) To preclude the mixing model, Ce/Pb is used in their modeling. Unfortunately, data quality of Pb analyses in different labs are variable and thus some abnormal Ce/Pb could be due to artificial effect. Nb is used in this study due to its immobility during weathering processes. Why not to use a ratio involving Nb, such a Nb/Th?

(e) Finally I feel uncomfortable to accept that a highly fusible PREMA can survive vigorous convection for billions of years. A highly fusible component can easily be molten in convection, and hence cannot be isolated from convection for long time and keep their primordial isotopic identity. How to reconcile this chemical and physical conflict in the interpretation remains ambiguous.

Response to Comments-Cai et al.

Reviewer #1 :

Dear Editor,

I had the pleasure of reading the manuscript by Cai et al., titled "Widespread PREMA in the Upper Mantle Indicated by Low-Degree Basaltic Melts", and I am genuinely excited by the work they have done. I believe they've made significant strides in their research, and their manuscript provides an insightful reading experience. However, I do have a few suggestions that could further enhance the clarity and depth of their study. We would like to thank Dr. Doucet for his positive attitude toward this work, including the detailed and constructive comments that have greatly helped us refine our arguments.

1. The authors have admirably updated the isotopic composition of PREMA. They've interestingly used Sr-Nd-Hf isotopes and have reported the $^{206}\text{Pb}/^{204}\text{Pb}$ value. However, I found myself wondering (a) why the values for $^{207}\text{Pb}/^{204}\text{Pb}$ and $^{208}\text{Pb}/^{204}\text{Pb}$ were not provided, and (b) whether Pb was actually utilized, or was it simply extrapolated from Sr-Nd-Hf. Given the focus on PREMA in this study, a detailed discussion on these points could provide valuable insights and clarify these aspects.

We did not include the $^{207}\text{Pb}/^{204}\text{Pb}$ and $^{208}\text{Pb}/^{204}\text{Pb}$ vs Nb-N plots in the original submission because they are similar to the $^{206}\text{Pb}/^{204}\text{Pb}$ vs Nb-N plot. For completeness, we have now provided these plots in the revised supplementary figures (see Fig. S8 and S9 which are also shown below). Although the ranges of Pb isotopes are wider than Sr-Nd-Hf isotopes, the low-Nb and high-Nb basalts show differences in Pb isotopes that resemble those exhibited by Sr-Nd-Hf isotopes, i.e., a large spread in the low-Nb basalts vs. Pb isotopic compositions typical of PREMA in the high-Nb basalts.

[Redacted]

[Redacted]

The Pb isotope ranges of PREMA in previous studies are variable, which may be due to the different ways used to define PREMA combined with less systematic Pb isotope variations in oceanic basalts than for Sr, Nd, and Hf isotopes. This is similarly noticeable in our data (Fig. S9). In addition, kimberlites do not have well-defined ranges in Pb isotope compositions (in part due to alteration susceptibility of U and the greater sensitivity to minor crustal contamination), preventing estimation of the kimberlite-based PREMA for Pb isotopes as clarified in lines 134-135 of the revised manuscript. The low Pb abundance in a relatively primitive mantle compared to enriched components, together with the larger fractionations of the U/Pb ratio during very minor melting, dehydration or re-enrichment mean that Pb isotopes will always be more sensitive to minor "processing" due to small-degree melting or re-addition of metasomatic melts. Finally, the more radiogenic Pb isotope composition of oceanic basalts, including PREMA, is difficult to reconcile with any simple mantle evolution

model, as recently shown by Doucet et al. (2023). To sum up, we believe that explaining the complexities of Pb isotopes goes beyond the scope of this manuscript.

2. Their rationale for using continental basalts (lines 76–79) could benefit from a bit more explanation. For instance, it would be great to understand why basalts need to be enriched in highly incompatible elements, and why a low degree of partial melting is preferable over a high degree. More elaborate reasoning here would undoubtedly strengthen their arguments.

Agreed - We have included this additional statement in lines 79-82 of the revised manuscript:

“As the signature of fusible components becomes diluted during high-degree melting of the mantle, and magmas with high trace element contents are not easily contaminated by crust, hence, these low-degree basalts are more useful to track the nature of highly fusible mantle components.”

3. The study opens up some intriguing questions about potential issues with continental basalts. The possibility of continental crust contamination is an aspect that could be examined, and it's not currently discussed in the manuscript. Moreover, it would be enlightening to explore the possibility of continental basalts being generated in the lithospheric mantle, as inferred by the author at <100km. Although these queries are addressed, they come a bit later in the manuscript and addressing them sooner could provide a smoother flow of ideas.

We agree that crustal contamination is an important issue to consider. As such, continental crust contamination was addressed by employing element ratios that are markedly different in oceanic basalts and continental crust (e.g., Nb/U, Ce/Pb; see Data compilation and filtering). The high-Nb basalts cannot be explained by crustal contamination because they have depleted Sr-Nd isotopes coupled with high Nb/U and Ce/Pb, with similar compositions observed in oceanic and continental magmas (Figure 3).

Moreover, the source of PREMA cannot be the lithospheric mantle because the PREMA component is ubiquitous in oceanic basalts and kimberlites (lines 148-150 in the revision), both of which feature sources deeper than the lithosphere. In addition, high-Nb basalts and kimberlites with a PREMA signature are associated with lithosphere of highly variable age and origin (and hence highly diverse isotopic composition), thus preventing a common lithospheric origin for PREMA (lines 157-158 in the revision). In some cases, asthenosphere-derived melts may generate metasomatic assemblages in the deep lithosphere which later, after short residence times, undergo melting to form continental basalts with asthenosphere signatures (Thompson et al., 2005; lines 159-163 in the revision). Yet, the ultimate mantle source of these geochemical signatures is the convecting mantle. This is because metasomatic veins cannot be preserved in the lithospheric mantle for a long time, their solidus being too low (Sun and Dasgupta, 2023).

4. Upon reviewing the provided data and engaging with the content, I am leaning

towards the idea that the PREMA signature could potentially stem from the lithosphere itself. This is a fascinating possibility that deserves further exploration.

This hypothesis has been addressed in Response to Comment 3. Some PREMA signatures may come from the very base of the lithosphere but they are short-lived features that are embedded there by asthenospheric melts with PREMA-like signatures (lines 159-163 in the revision).

The authors have indeed done a commendable job, and these suggestions are merely intended to help them enhance their work. I eagerly look forward to seeing the next iteration of their manuscript.

We hope that Dr. Doucet is satisfied with the manuscript revision.

Reviewer #2 (Anonymous):

This study address the origin and distribution of PREMA, by examing 3500 Cenozoic sodic continental and oceanic basalts. The authors show high-Nb basalts (equivalent to low degree of melts) tend to preferentially tap PREMA. The main conclusions of this study are: (a) The high-Nb basalts share a common source with kimberlites, being slightly higher degree of partial melts. (b) The PREMA does not result of mixing between depleted and enriched mantle components. (c) The PREMA is a physically discrete, highly fusible constituent that is widespread in Earth's upper mantle, rather than being exclusively associated with deep thermochemical structures such as LLSVPs. While this work is a potential contribution to NC, it has a number of weaknesses/problems that must be addressed before further consideration.

We are grateful to this reviewer for his/her constructive comments that have aided us to strengthen our arguments. We also hope this reviewer is satisfied with our explanations and efforts in the manuscript revision.

(a)High-Nb basalts are interpreted as low-degree melts, as such they are often alkalic in nature through and are associated with low Si content. In principle such melts are more abundant in continental setting than in oceanic setting, due to the lithospheric lid effect. While this is claimed in the text by quoting the proportion of 24% versus 7%, Fig. 1 does not show compelling evidence. For instance, Nb concentrations in the lavas from two settings are essentially similar. The continental basalts tend to be higher in Si at given Nb, which is the reverse of what is expected. In addition, the continental high-Nb basalts tend to have higher Sr but lower Nd isotopic ratios than oceanic counterparts (looking at the medium curves in Fig. 1 and individual data points). I am thus wondering whether they come from different sources and/or some assimilation processes were involved. Or this is related to the data processing which average out the compositional heterogeneity.

We thank the reviewer for the perceptive comment while noting that the reviewer has inadvertently misquoted “Fig. 3” as “Fig. 1”. We likely did not highlight the best plot to illustrate the point we were trying to make. Instead, Fig. S1b (reproduced below) shows the kernel density curves for Nb-N of MORB&BABBB (back-arc basin basalts), oceanic and continental basalts, and kimberlites. The differences in these curves are consistent with previous conclusions that lithospheric thickness dominates the degree of mantle melting and the Nb contents of melts (Niu et al.,2021 and Massuyeau et al., 2021). We reproduce a figure from Massuyeau et al. (2021) below to best illustrate this point.

[Redacted]

(The figure is from Massuyeau et al., 2021)

While the peak in kernel density distribution for continental basalts occurs at higher Nb-N than oceanic basalts, it is true that the difference is subtle. The subtle differences in Nb content between oceanic and continental settings may be due to the limited

differences in lid thickness. Because basalts, including nephelinites and melilitites, only occur in settings where the lithospheric lid is thinner than 125km (if thicker, mantle melts would be kimberlites or similar rocks), as shown in Fig 9 from Massuyeau et al. (2021; above), the difference in lithospheric thickness for melt production in continents and oceans is subtle if we only consider the lithosphere <125km for the basalt generation (Fig. 14bc from Zhu et al., 2021). Kimberlites, which erupt through thick cratons (normally ~150km or more) have distinctively higher Nb contents than basalts (see Fig. S1) as they represent even lower degrees of mantle partial melting.

[Redacted]

(The figure is from Zhu et al., 2021)

We agree that oceanic basalts show less scatter in Sr-Nd-Pb isotopes and SiO₂ contents than their continental counterparts at the same Nb contents. This may not be associated with the assimilation of continental crust because the Sr-Nd isotopes of high-Nb basalts from the continents do not show correlations with Nb/U- Ce/Pb (see figure below).

The most likely reason for the greater isotopic diversity in continental sodic basalts is that partial melting beneath the continental lithosphere taps additional heterogeneities from the longer-lived lithospheric mantle than oceanic basalts have access to. However, this potential contribution from the continental lithosphere does not affect our conclusion, because the trends/convergences of continental and oceanic basalts are similar and we focus on the basaltic melts with the highest Nb contents (devoid of continental lithosphere contamination) to best reflect the asthenosphere mantle source signature.

(b) Broadly similar trace element pattern is emphasized between high-Nb basalts and kimberlites. However, high-Nb basalts lack negative anomalies in Sr, Zr and Hf which

are conspicuous in Kimberlites. I am not convinced by an interpretation invoking different degree of partial melting. In this regard, a detail evaluation of source lithology is absolutely needed.

Negative anomalies of Zr-Hf of alkali basalts are directly related to the melting degree as approximated by Nb contents. Basalts with Nb-N > 150 show the most pronounced negative anomalies of Zr-Hf (as shown below). Conversely, negative anomalies for Zr-Hf are absent for elevated degrees of melting (i.e. Nb-N < 50). So the negative anomaly becomes more pronounced because of the greater enrichment in adjacent incompatible elements. The negative correlation between Zr/Nd and Nb contents supports this explanation (Fig. S5). Besides, the more obvious negative anomalies of Zr-Hf-Ti in kimberlites are associated with a higher proportion of garnet in the source ($D_{\text{garnet/carbonate-bearing melt}}$ of Zr-Hf-Ti > 1; Dasgupta et al., 2009), so degree of melting and residual mineralogy both play a role.

Similarly, conspicuous positive Sr anomalies in low-Nb basalts progressively disappear with decreasing extent of melting (Nb-N > 100) and become negative in kimberlites. Again, this is why the signatures that we focus on are cleanest in melts with Nb-N > 100. These variations can be understood based on the relative incompatibility of these elements during increasing degrees of mantle melting in the presence of CO₂ in the source and are completely compatible with the partitioning in these systems. We have added average trace element patterns of basalts with different Nb contents in Fig. 5a to hopefully make this clearer.

[Redacted]

(c) Classic PREMA is generally associated with high ³He/⁴He ratios which is considered as an intrinsic feature of PREMA. The so-called wide-spread PREMA in the upper mantle, as argued in this study, do not show this primordial He isotope composition. This observation may simply reflect different composition between the upper and lower mantle, arguing against widespread PREMA in the upper mantle.

We have to disagree with the reviewer here. Most mantle-derived melts with high ³He/⁴He have more geochemically depleted Sr-Nd-Pb isotope compositions than PREMA (Day et al., 2022; see figure below). Therefore, PREMA does not have a monopoly on high ³He/⁴He and the association of PREMA with high ³He/⁴He (Hart et al., 1992) is in fact, not strong – in keeping with our hypothesis. The core is also a potential source of high ³He/⁴He (Deng et al., 2023).

[Redacted]

(The figure is from Day et al., 2022)

(d) To preclude the mixing model, Ce/Pb is used in their modeling. Unfortunately, data quality of Pb analyses in different labs are variable and thus some abnormal Ce/Pb could be due to artificial effect. Nb is used in this study due to its immobility during weathering processes. Why not to use a ratio involving Nb, such a Nb/Th?

This is a good point that we have considered. We have undertaken mixing modeling using different ratios (e.g., Nb/U, Nb/Th, Nb/La, Ce/Pb, Zr/Nd, Ti/Gd). The results of

these plots are similar and show that mixing between DMM (Depleted MORB Mantle) and EM (Enriched Mantle) cannot generate high-Nb basalts. The plots have been added in supplementary materials (Figure S10) and enhance the argument. We thank the reviewer for this suggestion.

(e) Finally I feel uncomfortable to accept that a highly fusible PREMA can survive vigorous convection for billions of years. A highly fusible component can easily be molten in convection, and hence cannot be isolated from convection for long time and keep their primordial isotopic identity. How to reconcile this chemical and physical conflict in the interpretation remains ambiguous.

We understand the reviewer's concern but have to emphasize that even the most fusible mantle component can only melt at low pressure and even with 'vigorous' mantle convection cycling through melting zones is inefficient as we already demonstrated in geodynamic modeling (Extended Data Video 1). After 4.5 Gyr of evolution, our models show clearly that the average fraction of unprocessed mantle is about 25% (Fig. 7). Melting of carbon-bearing peridotites below 250-300 km is suppressed by low oxygen fugacities, within the stability field of metal saturation (Rohrbach et al., 2007; Rohrbach and Schmidt, 2011).

Nonetheless, many readers may share this concern and so to better address it we have included this additional statement in lines 231-235 of the revised manuscript: *“Survival of minimally processed and yet fusible PREMA in the convecting mantle for billions of years is consistent with inefficient cycling of mantle material through low-pressure melting zones (Extended Data Video 1) as well as limited melting of carbon-bearing peridotites at the prevailing reducing conditions that occur below the metal saturation depth (~250km).”*

References

- Dasgupta, R., Hirschmann, M. M., McDonough, W. F., Spiegelman, M. & Withers, A. C. Trace element partitioning between garnet lherzolite and carbonatite at 6.6 and 8.6 GPa with applications to the geochemistry of the mantle and of mantle-derived melts. *Chem. Geol.* 262, 57–77 (2009).
- Day, J. M. D., Jones, T. D. & Nicklas, R. W. Mantle sources of ocean islands basalts revealed from noble gas isotope systematics. *Chem. Geol.* 587, 120626 (2022).
- Deng, J., Du, Z. Primordial helium extracted from the Earth's core through magnesium oxide exsolution. *Nat. Geosci.* 16, 541–545 (2023)
- Doucet, L. S. et al. The global lead isotope system: Toward a new framework reflecting Earth's dynamic evolution. *Earth-Science Rev.* 243, 104483 (2023).
- Hart, S. R., Hauri, E. H., Oschmann, L. A. & Whitehead, J. A. Mantle plumes and entrapment: Isotopic evidence. *Science* 256, 517–520 (1992).
- Massuyeau, M. et al. MAGLAB: A computing platform connecting geophysical signatures to melting processes in Earth's mantle. *Phys. Earth Planet. Inter.* 314, (2021).
- Niu, Y. Lithosphere thickness controls the extent of mantle melting, depth of melt extraction and basalt compositions in all tectonic settings on Earth – A review and new perspectives. *Earth-Science Reviews* vol. 217 (2021).

- Rohrbach, A. & Schmidt, M. W. Redox freezing and melting in the Earth's deep mantle resulting from carbon-iron redox coupling. *Nature* 472, 209–214 (2011).
- Rohrbach, A. et al. Metal saturation in the upper mantle. *Nature* 449, 456–458 (2007).
- Sun, C. & Dasgupta, R. Carbon budget of Earth's deep mantle constrained by petrogenesis of silica-poor ocean island basalts. *Earth Planet. Sci. Lett.* In Press, 118135 (2023).
- Thompson, R. N. et al. Source of the quaternary alkalic basalts, picrites and basanites of the Potrillo Volcanic Field, New Mexico, USA: Lithosphere or convecting mantle? *J. Petrol.* 46, 1603–1643 (2005).
- Zhu, R., Zhao, G., Xiao, W., Chen, L., & Tang, Y. (2021). Origin, accretion, and reworking of continents. *Reviews of Geophysics*, 59, e2019RG000689.

REVIEWER COMMENTS

Reviewer #1 (Remarks to the Author):

Dear Rebecca,

I have reviewed both the revised manuscript and authors responses to the comments I made on the previous version of the manuscript. I must say, I'm quite pleased with the amendments and explanations the authors provided, which have effectively addressed my initial, minor concerns.

Reviewer #2 (Remarks to the Author):

The authors have done a formidable job to address nearly all the concerns raised by reviewers. While I think their rebuttal clarified a number of ambiguities in the previous version, some concerns still persisted, as I am not convinced yet.

(1)The authors agree with my previous remark that continental basalts show larger scatters in Sr-Nd-Pb isotopes and SiO₂ contents than ocean counterparts. They claimed that this is not related to the crustal assimilation because the Sr-Nd isotopes of high-Nb basalts from the continents do not show correlations with Nb/U- Ce/Pb. However, looking at the figure in Page 8 of the rebuttal letter, I saw a broadly negative correlation between Ce/Pb and $^{87}\text{Sr}/^{86}\text{Sr}$, that is consistent with assimilation. On the other hand, the authors did not address the remark of systematically higher SiO₂ contents (not just larger scatter only) of continental basalts at comparable Nb contents compared with their oceanic counterparts.

(2)There are no obvious correlations in other plots. However this cannot be used as compelling evidence in ruling out the assimilation possibility, a problem simply associated with the method of using global data. For instance, no correlation exists apparently in the $^{87}\text{Sr}/^{86}\text{Sr}$ ratio against SiO₂. Be aware that the range of SiO₂ is 40-50% (10%) which is considerable making such a plot no sense in discriminating assimilation vs source, because global data would mask any trends defined by individual suites. Specifically, the samples with the lowest $^{87}\text{Sr}/^{86}\text{Sr}$ have a narrow SiO₂ range, those with 0.704 exhibit a much a larger Si range, and the highest $^{87}\text{Sr}/^{86}\text{Sr}$ are only associated with the samples with high Si (no single sample with low Si is noticed). I have no idea how to explain this without assimilation. Finally, I have to indicate that assimilation referred here is a general definition, meaning crustal components could resident either in crust or in the mantle.

(3)I have my reservation on their arguments on the He isotopic issue.

Response to Comments-Cai et al.

Reviewer #1:

I have reviewed both the revised manuscript and authors responses to the comments I made on the previous version of the manuscript. I must say, I'm quite pleased with the amendments and explanations the authors provided, which have effectively addressed my initial, minor concerns.

We would like to thank Dr. Doucet again for his appreciation and support of this work.

Reviewer #2 (Anonymous):

The authors have done a formidable job to address nearly all the concerns raised by reviewers. While I think their rebuttal clarified a number of ambiguities in the previous version, some concerns still persisted, as I am not convinced yet.

We would like to thank this reviewer for their constructive criticism and hope this further revision will be considered satisfactory.

(1)The authors agree with my previous remark that continental basalts show larger scatters in Sr-Nd-Pb isotopes and SiO₂ contents than ocean counterparts. They claimed that this is not related to the crustal assimilation because the Sr-Nd isotopes of high-Nb basalts from the continents do not show correlations with Nb/U- Ce/Pb. However, looking at the figure in Page 8 of the rebuttal letter, I saw a broadly negative correlation between Ce/Pb and ⁸⁷Sr/⁸⁶Sr, that is consistent with assimilation. On the other hand, the authors did not address the remark of systematically higher SiO₂ contents (not just larger scatter only) of continental basalts at comparable Nb contents compared with their oceanic counterparts.

(2)There are no obvious correlations in other plots. However this cannot be used as compelling evidence in ruling out the assimilation possibility, a problem simply associated with the method of using global data. For instance, no correlation exists apparently in the ⁸⁷Sr/⁸⁶Sr ratio against SiO₂. Be aware that the range of SiO₂ is 40-50% (10%) which is considerable making such a plot no sense in discriminating assimilation vs source, because global data would mask any trends defined by individual suites. Specifically, the samples with the lowest ⁸⁷Sr/⁸⁶Sr have a narrow SiO₂ range, those with 0.704 exhibit a much a larger Si range, and the highest ⁸⁷Sr/⁸⁶Sr are only associated with the samples with high Si (no single sample with low Si is noticed). I have no idea how to explain this without assimilation. Finally, I have to indicate that assimilation referred here is a general definition, meaning crustal components could resident either in crust or in the mantle.

The above two comments are both about potential crustal assimilation in continental basalts. Hence, we address them together.

While we don't fully agree with the above assessment of the trends observed by the reviewer (e.g., we cannot derive any statistically robust correlation between $^{87}\text{Sr}/^{86}\text{Sr}$ and Ce/Pb in continental basalts), we concede that the higher SiO_2 contents at a given Nb content in continental basalts compared to oceanic basalts might be attributed to a lithospheric (including crust) contribution. This point is now stated clearly in the Results section (lines 115-119). However, we note the trends/convergences of radiogenic isotope ratios (e.g., Sr-Nd-Hf isotopes) and elemental ratios (e.g., Nb/Th, Ce/Pb, Ti/Eu) with increasing Nb contents occur in both continental and oceanic basalts. Therefore, while the potential contribution from continental lithosphere provides some additional scatter, perhaps most noticeably in the SiO_2 contents of continental sodic basalts, the compositions of continental and oceanic basalts with high Nb contents reflect broadly similar asthenospheric mantle sources (lines 119-125). We hope that these changes are satisfactory and note that while the comments are valid and well-taken, our consideration of them do not affect the conclusions of this manuscript.

(3) I have my reservation on their arguments on the He isotopic issue.

The reviewer states that s/he has 'reservation' on our arguments on the He isotopic data, but without specific objective statement about what the reservation is, it is impossible for us to respond to this comment. We have therefore not made any modifications in the text.

REVIEWERS' COMMENTS

Reviewer #2 (Remarks to the Author):

I have no more comments on the manuscript, which is ready to be accepted

Third Round of Review

REVIEWERS' COMMENTS

Reviewer #2 (Anonymous):

I have no more comments on the manuscript, which is ready to be accepted.

RESPONSE: Thanks.